# A Conditional Mutual Information Estimator for Mixed Data and an Associated Conditional Independence Test

**DOI:** 10.3390/e24091234

**Published:** 2022-09-02

**Authors:** Lei Zan, Anouar Meynaoui, Charles K. Assaad, Emilie Devijver, Eric Gaussier

**Affiliations:** 1Department of Mathematics, Information and Communication Sciences, Université Grenoble Alpes, CNRS, Grenoble INP, LIG, 38000 Grenoble, France; 2R&D Department, EasyVista, 38000 Grenoble, France

**Keywords:** mixed data, conditional mutual information, conditional independence testing, permutation tests

## Abstract

In this study, we focus on mixed data which are either observations of univariate random variables which can be quantitative or qualitative, or observations of multivariate random variables such that each variable can include both quantitative and qualitative components. We first propose a novel method, called CMIh, to estimate conditional mutual information taking advantages of the previously proposed approaches for qualitative and quantitative data. We then introduce a new local permutation test, called LocAT for local adaptive test, which is well adapted to mixed data. Our experiments illustrate the good behaviour of CMIh and LocAT, and show their respective abilities to accurately estimate conditional mutual information and to detect conditional (in)dependence for mixed data.

## 1. Introduction

Measuring the (in)dependence between random variables from data when the underlying joint distribution is unknown plays a key role in several settings, as in causal discovery [1], graphical model inference [2] or feature selection [3]. Many dependence measures have been introduced in the literature to quantify the dependence between random variables, as *Mutual Information* (MI) [4], *distance correlation* [5], kernel-based measures such as the *Hilbert–Schmidt Independence Criterion* (HSIC) [6], *COnstrained COvariance* (COCO) [7] or copula-based approaches [8]. We focus in this work on (conditional) mutual information, which has been successfully used in various contexts and has shown good practical performance in terms of the statistical power of the associated independence tests [9], and consider both quantitative and qualitative variables. A quantitative variable is a variable which has infinite support and values on which one can use more complex distances than the mere (0−D) distance (which is 0 for two identical points and D for points with different values). All variables which do not satisfy these two conditions are deemed qualitative. Note that one can use the (0−D) distance on any type of variables, and that this distance is the standard distance for nominal variables; one can of course use, if they exist, other distances than the (0−D) on qualitative variables. Continuous variables as well as ordinal variables with infinite support are here quantitative, whereas nominal variables and ordinal variables with finite support are considered qualitative.

The conditional mutual information [10] between two quantitative random variables *X* and *Y* conditionally to a quantitative random variable *Z* is given by: (1)$$I\left(X;Y|Z\right)=\;\int \int \int \;P_{XYZ}\left(x,y,z\right)\log\left(\frac{P_{XY|Z}\left(x,y|z\right)}{P_{X|Z}\left(x|z\right)P_{Y|Z}\left(y|z\right)}\right)\;dxdydz,$$
where $P_{XYZ}$ is the joint density of (X,Y,Z) and $P_{XY|Z}$ (respectively, $P_{X|Z}$ and $P_{Y|Z}$) is the density of (X,Y) (respectively, *X* and *Y*) given *Z*. Note that Equation (Equation 1) also applies to qualitative variables by replacing integrals by sums and densities by mass functions. The conditional mutual information can also be expressed in terms of entropies as:$$\begin{array}{c} I(X;Y|Z)=H(X,Z)+H(Y,Z)-H(X,Y,Z)-H(Z),\; \end{array}$$
where H(·) is the Shannon entropy [11] defined as follows for a quantitative random variable *X* with density $P_{X}$:$$H\left(X\right)=-\int P_{X}\left(x\right)\log\left(P_{X}\left(x\right)\right)\;dx.\;$$

Conditional mutual information characterizes conditional independence in the sense that I(X;Y|Z)=0 if and only if *X* and *Y* are independent conditionally to *Z*.

Estimating conditional mutual information for purely qualitative or purely quantitative random variables is a well-studied problem [12,13]. The case of mixed datasets comprising both quantitative and qualitative variables is, however, less studied even though mixed data are present in many applications, for example as soon as one needs to threshold some quantitative values as in monitoring systems. The aim of this paper is to present a new statistical method to detect conditional (in)dependence for mixed data. To do so, we introduce both a new estimator of conditional mutual information as well as a new test to conclude on the conditional (in)dependence.

The remainder of the paper is organized as follows. Section 2 describes related work. We introduce in Section 3 our estimator, as well as some numerical comparisons with existing conditional mutual information estimators. We present in Section 4 the associated independence tests as well as numerical studies conducted on simulated and real datasets. Finally, Section 5 concludes the paper.

## 2. Related Work

We review here related work on (conditional) mutual information estimators as well as on conditional independence testing.

### 2.1. Conditional Mutual Information

A standard approach to estimate (conditional) mutual information from mixed data is to discretize the data and to approximate the distribution of the random variables by a histogram model defined on a set of intervals called bins [14]. Each bin corresponds to a single point for qualitative variables and to consecutive non-overlapping intervals for quantitative variables. Even if the approximation becomes better when dealing with smaller bins, finite sample size requires to carefully choose the number of bins. To efficiently generate adaptive histograms model from quantitative variables, Cabeli et al. [15] and Marx et al. [16] transform the problem into a model selection problem, using a criterion based on the minimum description length (MDL) principle. An iterative greedy algorithm is proposed to obtain the histogram model that minimizes the MDL score, from which one can derive joint and marginal distributions. The difference between the two methods rely on the estimation, which is conducted for each entropy term in Cabeli et al. [15] and globally in Marx et al. [16]. These approaches are very precise to estimate the value of the (conditional) mutual information even in multi-dimensional cases, but are computational costly when the dimensions increase.

To estimate entropy, two main families of approaches have been proposed. The first one is based on kernel-density estimates [17] and applies to quantitative data, whereas the second one is based on *k*-nearest neighbours and applies to both qualitative and quantitative data. The second one is preferred as it naturally adapts to the data density and does not require extensive tuning of kernel bandwidths. Using nearest neighbours of observations to estimate the entropy dates back to Kozachenko and Leonenko [18], which was then generalized to a *k*-nearest neighbour (kNN) approach by Singh et al. [19]. In this method, the distance to the $k^{th}$ nearest neighbour is measured for each data point, the probability density around each data point being substituted into the entropy expression. When *k* is fixed and the number of points is finite, each entropy term is noisy and the estimator is biased. However, this bias is distribution independent and can be subtracted out [20]. Along this line, Kraskov et al. [21] proposed an estimator for mutual information that goes beyond the sum of entropy estimators. This latter work was then extended to conditional mutual information in Frenzel and Pompe [12]. The resulting model, called FP, however, only deals with quantitative data.

More recently, Ross [22] and Gao et al. [23] introduced two approaches to estimate mutual information for mixed data, however, without any conditioning set. Following these studies, Rahimzamani et al. [24] proposed a measure of incompatibility between the joint probability $P_{XYZ}\left(x,y,z\right)$ and its factorization $P_{X|Z}\left(x|z\right)P_{Y|Z}\left(y|z\right)P_{Z}\left(z\right)$ called *graph divergence measure* and extended the estimator proposed in Gao et al. [23] to conditional mutual information, leading to a method called RAVK. As ties can occur with a non zero probability in mixed data, the number of neighbours has to be carefully chosen. Even more recently, Mesner and Shalizi [25] extended FP [12] to the mixed data case by introducing a qualitative distance metric for non-quantitative variables, leading to a method called MS. The choice of the qualitative and quantitative distances is a crucial point in MS [26]. FP, RAVK and MS all lead to an estimator of the form:$$\hat{I}\left(X;Y|Z\right)=\frac{1}{n}\sum_{i=1}^{n}\psi \left(k_{Ge,i}\right)-f\left(n_{Ge,XZ,i}\right)-f\left(n_{Ge,YZ,i}\right)+f\left(n_{Ge,Z,i}\right).\;$$
Ge stands for either FP, RAVK or MS, *n* represents the number of the observations and ψ(.) is the digamma function. For FP, $k_{FP,i}$ is a constant hyper-parameter, $f\left(n_{FP,W,i}\right)=\psi \left(n_{FP,W,i}+1\right)$ with *W* being either (X,Z), (Y,Z) or (Z). Denoting, as usual, the $\ell_{\infty }$ distance between *i* and its *k*-nearest-neighbour in global space by $\rho_{k,i}/2$, $n_{FP,W,i}$ represents the number of points in the joint space *W* that have an $\ell_{\infty }$ distance strictly smaller than $\rho_{k,i}/2$:(2)$$n_{FP,W,i}=\left|\left\{j:||\omega_{i}-\omega_{j}{\left|\right|}_{\infty }<\rho_{k,i}/2,j\neq i\right\}\right|,\;$$
where $\omega_{j}$ and $\omega_{i}$ represent the coordinates of the points in the space corresponding to *W*. For RAVK, to adapt it to mixed data, Mesner and Shalizi [25] proposed the use of $f\left(n_{RAVK,W,i}\right)=\log\left(n_{RAVK,W,i}+1\right)$ where $n_{RAVK,W,i}$ includes boundary points:(3)$$n_{RAVK,W,i}=\left|\left\{j:||\omega_{i}-\omega_{j}{\left|\right|}_{\infty }\leq \rho_{k,i}/2,j\neq i\right\}\right|.\;$$
Furthermore, $k_{RAVK,i}=n_{RAVK,(X,Y,Z),i}$. Lastly, for MS, one has $f\left(n_{MS,W,i}\right)=\psi \left(n_{MS,W,i}\right)$, $n_{MS,W,i}$ and $k_{MS,i}$ being defined as for RAVK.

We also want to mention the proposal made by Mukherjee et al. [27] of a two-stage estimator based on generative models and classifiers as well as the refinement introduced in Mondal et al. [28] and based on a neural network that integrates the two stages into a single training process. It is, however, not clear how to adapt to mixed data these methods primarily developed for quantitative data.

### 2.2. Conditional Independence Tests

To decide whether the estimated conditional mutual information value is small enough to conclude on the (in)dependence of two variables *X* and *Y* conditionally to a third variable *Z* in a finite sample regime, one usually relies on statistical independence tests. The null and the alternative hypotheses are, respectively, defined by
H0:X⊥⊥Y|ZandH1:X⊥⊥Y|Z,
where ⊥⊥ means *independent of* and ⊥⊥ means *not independent of*. In the independence testing literature, two main families exist, asymptotic and non-asymptotic ones (see e.g., [29], chapter 3). The former is used when the sample size is big enough and relies on the asymptotic distribution of the estimator under the null hypothesis, while the latter applies to any sample size without any prior knowledge of the asymptotic null distribution of the estimator. Note that conditional independence testing is a more difficult problem than its non-conditional counterpart [30]. In particular, the asymptotic behaviour of the conditional mutual information estimator under the null hypothesis is usually unknown.

Kernel-based tests are known for their capability to deal with nonlinearity and high dimensions. The Hilbert–Schmidt independence criterion (HSIC) has been first proposed for testing unconditional independence. Fukumizu et al. [31] extended HSIC to the conditional independence setting using the Hilbert-Schmidt norm of the conditional cross-covariance operator. Another representative of this test category is the kernel conditional independence test (KCIT) proposed by Zhang et al. [32]. It works by testing for vanishing correlation between residual functions in reproducing kernel Hilbert spaces. To reduce the computational complexity of KCIT, Strobl et al. [33] used random Fourier features to approximate KCIT and thereby proposed two tests, namely the randomised conditional independence test that explores the partial cross-covariance matrix between (X,Z) and *Y*, and the randomized conditional correlation test (RCoT) that tests *X* and *Y* after some transformations to remove the effect of *Z*. RCoT can be related to two-step conditional independence testing [34], computing first conditional expectations of feature maps and then testing the residuals. Doran et al. [35] also proposed a kernel conditional independence permutation test. They used a specific permutation of the samples to generate data from $P_{X|Z}\left(x|z\right)P_{Y|Z}\left(y|z\right)P_{Z}\left(z\right)$ which unfortunately requires solving a time-consuming linear program, then performed a kernel-based two-sample test [36]. However, kernel-based tests need to carefully adjust bandwidth parameters that characterise the length scales in the different subspaces of X,Y,Z and can only be implemented on purely quantitative data.

More recently, Shah and Peters [30] proposed the generalised covariance measure (GCM) test. For univariate *X* and *Y*, instead of testing for independence between the residuals from regressing *X* and *Y* on *Z*, the GCM tests for vanishing correlations. How to extend this approach to mixed data is, however, not clear. Tsagris et al. [37] employed likelihood-ratio tests based on regression models to devise conditional independence tests for mixed data; however, in their approach one needs to postulate a regression model.

Permutation tests [38] are popular when one wants to avoid assumptions on the data distribution. For testing the independence of *X* and *Y* conditionally to *Z*, permutation tests randomly permute all values in *X*. If this destroys the potential dependence between *X* and *Y*, as desired, this also destroys the one between *X* and *Z*, which is not desirable. In order to preserve the dependence between *X* and *Z*, Runge [39] proposed a local permutation test in which permutations within *X* are conducted within similar values of *Z*. We extend in this paper this test, designed for quantitative data, to the mixed data case.

## 3. Hybrid Conditional Mutual Information Estimation for Mixed Data

The two most popular approaches to estimate conditional mutual information are based on the *k*-nearest neighbour method [12,21], which has been mostly used on quantitative variables, or on histograms [15,16], particularly adapted to qualitative variables. We show in this section how these two approaches can be combined to derive an estimator for mixed data.

Let us consider three mixed random vectors *X*, *Y* and *Z*, where any of their components can be either qualitative or quantitative. Let us denote by $X^{t}$ (respectively, $Y^{t}$, $Z^{t}$) the sub-vector of *X* (respectively, *Y*, *Z*) composed by the quantitative components. Similarly, we denote by $X^{\ell }$ (respectively, $Y^{\ell }$, $Z^{\ell }$)the sub-vector of qualitative components of *X* (respectively, *Y*, *Z*). Then, from the permutation invariance property of Shannon entropy, the conditional mutual information can be written as:$$\begin{array}{cc} I(X;Y|Z)= & H\left(X^{},Z^{}\right)+H\left(Y^{},Z^{}\right)-H\left(X^{},Y^{},Z^{}\right)-H\left(Z^{}\right)\\ = & H\left(X^{t},X^{\ell },Z^{t},Z^{\ell }\right)+H\left(Y^{t},Y^{\ell },Z^{t},Z^{\ell }\right)-H\left(X^{t},X^{\ell },Y^{t},Y^{\ell },Z^{t},Z^{\ell }\right)-H\left(Z^{t},Z^{\ell }\right). \end{array}$$

Now, from the property H(U,V)=H(U)+H(V|U), which is valid for any couple of random variables (U,V), one gets:(4)$$\begin{array}{cc} I(X;Y|Z)= & H\left(X^{t},Z^{t}|X^{\ell },Z^{\ell }\right)+H\left(Y^{t},Z^{t}|Y^{\ell },Z^{\ell }\right)-H\left(X^{t},Y^{t},Z^{t}|X^{\ell },Y^{\ell },Z^{\ell }\right)\\ \; & -H\left(Z^{t}|Z^{\ell }\right)+H\left(X^{\ell },Z^{\ell }\right)+H\left(Y^{\ell },Z^{\ell }\right)-H\left(X^{\ell },Y^{\ell },Z^{\ell }\right)-H\left(Z^{\ell }\right). \end{array}$$
Note that here the conditioning is only expressed with respect to qualitative components, which leads to a simpler estimation than the one obtained by conditioning with quantitative variables. We now detail how the different terms in the above expression are estimated.

### 3.1. Proposed Hybrid Estimator

Let us now consider an independently and identically distributed sample of size *n* denoted ${\left(X_{i},Y_{i},Z_{i}\right)}_{i=1,\ldots ,n}$. We estimate the qualitative entropy terms of Equation (Equation 4), namely $H(X^{\ell },Z^{\ell })$, $H(Y^{\ell },Z^{\ell })$, $H(X^{\ell },Y^{\ell },Z^{\ell })$ and $H(Z^{\ell })$, using histograms in which bins are defined by the Cartesian product of qualitative values. We provide here the estimation of $H(X^{\ell },Z^{\ell })$, the other terms are estimated in the same way. The theoretical entropy is expressed as: $$\begin{array}{cc} H(X^{\ell },Z^{\ell }) & =-E\left[\log P_{X^{\ell },Z^{\ell }}\left(X^{\ell },Z^{\ell }\right)\right]=-\sum_{\begin{array}{c} x^{\ell }\in \Omega \left(X^{\ell }\right)\\ z^{\ell }\in \Omega \left(Z^{\ell }\right) \end{array}}P_{X^{\ell },Z^{\ell }}\left(x^{\ell },z^{\ell }\right)\log\left(P_{X^{\ell },Z^{\ell }}\left(x^{\ell },z^{\ell }\right)\right),\; \end{array}$$
where Ω(·) corresponds to the probability space of a given random variable and $P_{X^{\ell },Z^{\ell }}$ is the probability distribution of $\left(X^{\ell },Z^{\ell }\right)$. The probability distribution of qualitative variables can be directly estimated via their empirical versions:(5)$${\hat{P}}_{X^{\ell },Z^{\ell }}\left(x^{\ell },z^{\ell }\right)=\frac{1}{n}\sum_{i=1}^{n}{𝟙}_{\left\{\left(X_{i}^{\ell },Z_{i}^{\ell }\right)=\left(x^{\ell },z^{\ell }\right)\right\}},\;$$
with ${𝟙}_{\left\{\cdot \right\}}$ is the indicator function. The resulting plug-in estimator is then given by
(6)$$\hat{H}\left(X^{\ell },Z^{\ell }\right)=-\sum_{\begin{array}{c} x^{\ell }\in \Omega \left(X^{\ell }\right)\\ z^{\ell }\in \Omega \left(Z^{\ell }\right) \end{array}}{\hat{P}}_{X^{\ell },Z^{\ell }}\left(x^{\ell },z^{\ell }\right)\log\left({\hat{P}}_{X^{\ell },Z^{\ell }}\left(x^{\ell },z^{\ell }\right)\right).\;$$

Let us now turn to the conditional entropies of Equation (Equation 4) for quantitative variables conditioned on qualitative variables and let us consider the term $H(X^{t},Z^{t}|X^{\ell },Z^{\ell })$. By marginalizing on $\left(X^{\ell },Z^{\ell }\right)$ one obtains:(7)$$H\left(X^{t},Z^{t}|X^{\ell },Z^{\ell }\right)=\sum_{\begin{array}{c} x^{\ell }\in \Omega \left(X^{\ell }\right)\\ z^{\ell }\in \Omega \left(Z^{\ell }\right) \end{array}}H\left(X^{t},Z^{t}|X^{\ell }=x^{\ell },Z^{\ell }=z^{\ell }\right)P_{X^{\ell },Z^{\ell }}\left(x^{\ell },z^{\ell }\right).\;$$
As before, the probabilities involved in Equation (Equation 7) are estimated by their empirical versions. The estimation of the conditional entropies $H(X^{t},Z^{t}|X^{\ell }=x^{\ell },Z^{\ell }=z^{\ell })$ is performed using the classical nearest neighbour estimator [19] with the constraint that $\left(X^{\ell },Z^{\ell }\right)=\left(x^{\ell },z^{\ell }\right)$: the estimation set consists of the sample points such that $\left(X^{\ell },Z^{\ell }\right)=\left(x^{\ell },z^{\ell }\right)$. The resulting estimator is given by:(8)$$\hat{H}\left(X^{t},Z^{t}|X^{\ell }=x^{\ell },Z^{\ell }=z^{\ell }\right)=\psi \left(n_{xz}\right)-\psi \left(k_{xz}\right)+\log\left(v_{d_{xz}}\right)+\frac{d_{xz}}{n_{xz}}\sum_{i=1}^{n_{xz}}\log\xi_{xz}\left(i\right),\;$$
where ψ is the digamma function, $n_{xz}$ is the size of the subsample space for which $\left(X_{i}^{\ell },Z_{i}^{\ell }\right)=\left(x^{\ell },z^{\ell }\right)$, $\xi_{xz}\left(i\right)$ is twice the distance of the $i^{th}$ subsample point to its $k_{xz}$ nearest neighbour, and $k_{xz}$ is the number of nearest neighbours retained. In the sequel, we set $k_{xz}$ to $\max(\left\lfloor n_{xz}/10\right\rfloor ,1)$, with ⌊·⌋ the floor function, following Runge [39] which showed that this value behaves well in practice. As originally proposed in [21] and adopted in subsequent studies, we rely on the $\ell_{\infty }$-distance which is associated with the maximum norm: for a vector $w=(w_{1},\ldots ,w_{m})$ in $R^{m}$, ${∥w∥}_{\infty }=\max(|w_{1}\left|,\ldots ,\right|w_{m}\left|\right)$. Finally, $d_{xz}$ is the dimension of the vector $\left(X^{t},Z^{t}\right)$ and $v_{d_{xz}}$ is the volume of the unit ball for the distance metric associated with the maximum norm in the joint space associated with $X^{t}$ and $Z^{t}$. The other entropy terms are estimated in the same way, the associated estimators being denoted by $\hat{H}\left(Z^{t}|Z^{\ell }\right)$, $\hat{H}\left(Y^{t},Z^{t}|Y^{\ell },Z^{\ell }\right)$ and $\hat{H}\left(X^{t},Y^{t},Z^{t}|X^{\ell },Y^{\ell },Z^{\ell }\right)$.

The conditional mutual information estimator for mixed data, which we will refer to as *CMIh*, finally amounts to:(9)$$\begin{array}{cc} \hat{I}\left(X;Y|Z\right) & =\hat{H}\left(X^{t},Z^{t}|X^{\ell },Z^{\ell }\right)+\hat{H}\left(Y^{t},Z^{t}|Y^{\ell },Z^{\ell }\right)-\hat{H}\left(X^{t},Y^{t},Z^{t}|X^{\ell },Y^{\ell },Z^{\ell }\right)\\ \; & -\hat{H}\left(Z^{t}|Z^{\ell }\right)+\hat{H}\left(X^{\ell },Z^{\ell }\right)+\hat{H}\left(Y^{\ell },Z^{\ell }\right)-\hat{H}\left(X^{\ell },Y^{\ell },Z^{\ell }\right)-\hat{H}\left(Z^{\ell }\right), \end{array}$$
where the different terms are obtained through Equations (Equation 5)–(Equation 8). Notice that all the volume-type terms, as for the $\log(v_{d_{xz}})$ term in Equation (Equation 8), are canceled out in Equation (Equation 9). Indeed, it is well known that the volume of the unit ball in $R^{p}$ with respect to to the maximum norm is $2^{p}$ and this leads to the following plain equation:$$\log\left(v_{d_{xyz}}\right)-\log\left(v_{d_{xz}}\right)-\log\left(v_{d_{yz}}\right)+\log\left(v_{d_{z}}\right)=\log\left(\frac{2^{d_{xyz}}2^{d_{z}}}{2^{d_{xz}}2^{d_{yz}}}\right)=\log\left(\frac{2^{d_{x}+d_{y}+d_{z}}2^{d_{z}}}{2^{d_{x}+d_{y}}2^{d_{y}+d_{z}}}\right)=0.$$

The main steps to derive the hybrid estimator CMIh are summarized in Algorithm 1.
**Algorithm 1** Hybrid estimator CMIh**Input**${\left(X_{i},Y_{i},Z_{i}\right)}_{i=1,\ldots ,n}$ the data, isCat indexes of qualitative components;Separate qualitative and quantitative components ${\left(X_{i}^{t},X_{i}^{\ell },Y_{i}^{t},Y_{i}^{\ell },Z_{i}^{t},Z_{i}^{\ell }\right)}_{i=1,\ldots ,n}$ using isCat;pointsInBin={}: indexes of points in each bin;densityOfBin={}: frequency of each bin;qualitativeEntropy=0: entropy of qualitative components;quantitativeEntropy=0: entropy of quantitative components;**if**$(X^{\ell },Y^{\ell },Z^{\ell })=\emptyset$**then**    qualitativeEntropy+=0;    **if** $(X^{t},Y^{t},Z^{t})=\emptyset$ **then**        quantitativeEntropy+=0;    **else**        Compute $\hat{H}\left(X^{t},Y^{t},Z^{t}\right)$ using the analogous of Equation (Equation 8);        $quantitativeEntropy+=\hat{H}\left(X^{t},Y^{t},Z^{t}\right)$;    **end if****else**    **for** $\left(x^{\ell },y^{\ell },z^{\ell }\right)\in \Omega \left(\left(X^{\ell },Y^{\ell },Z^{\ell }\right)\right)$ **do**        $pointsInBin\left[\left(x^{\ell },y^{\ell },z^{\ell }\right)\right]=\left\{i\in \left\{1,\cdots ,n\right\}:\left(X_{i}^{\ell }=x^{\ell },Y_{i}^{\ell }=y^{\ell },Z_{i}^{\ell }=z^{\ell }\right)\right\}$;        $densityOfBin\left[\left(x^{\ell },y^{\ell },z^{\ell }\right)\right]=length\left(pointsInBin\left[\left(x^{\ell },y^{\ell },z^{\ell }\right)\right]\right)/n$;    **end for**    $\hat{H}\left(X^{\ell },Y^{\ell },Z^{\ell }\right)=0$;    **for** k∈keys(densityOfBin) **do**        p=densityOfBin[k];        $\hat{H}\left(X^{\ell },Y^{\ell },Z^{\ell }\right)+=-p\log\left(p\right)$;    **end for**    $qualitativeEntropy+=\hat{H}\left(X^{\ell },Y^{\ell },Z^{\ell }\right)$;    **if** $(X^{t},Y^{t},Z^{t})=\emptyset$ **then**        quantitativeEntropy+=0;    **else**        **for** k∈keys(densityOfBin) **do**           p=densityOfBin[k];           Compute $\hat{H}\left(X^{t},Y^{t},Z^{t}|\left(X^{\ell },Y^{\ell },Z^{\ell }\right)=k\right)$ using the analogous of Equation (Equation 8) on observations pointsInBin[k];           $\hat{H}\left(X^{t},Y^{t},Z^{t}|X^{\ell },Y^{\ell },Z^{\ell }\right)+=p\hat{H}\left(X^{t},Y^{t},Z^{t}|\left(X^{\ell },Y^{\ell },Z^{\ell }\right)=k\right)$;        **end for**        $quantitativeEntropy+=\hat{H}\left(X^{t},Y^{t},Z^{t}|X^{\ell },Y^{\ell },Z^{\ell }\right)$;    **end if****end if**Compute other terms in Equation (Equation 9) using marginalization of the joint density;$\hat{I}\left(X;Y|Z\right)=quantitativeEntropy+qualitativeEntropy$;**Output**$\hat{I}\left(X;Y|Z\right)$.

**Remark** **1.**
*It is worth mentioning that our estimation of the entropy of the quantitative part is slightly different from the one usually used. In our estimation, the choice of the number of nearest neighbours is conducted independently for each entropy term and only with respect to the corresponding subsample size. This methodological choice yields more accurate estimators. Another important point is that the nearest neighbours are always computed on quantitative components as the qualitative components serve only as conditioning in Equation (Equation 9) or are involved in entropy terms estimated through Equation (Equation 6). Because of that, we can dispense with defining a distance on qualitative components, which is tricky as illustrated in Section 3.2.*


*Consistency*. Interestingly, the above hybrid estimator is asymptotically unbiased and consistent, as shown below.

**Theorem** **1.**
*Let (X,Y,Z) be a qualitative-quantitative mixed random vector. The estimator $\hat{I}\left(X;Y|Z\right)$ defined in Equation (Equation 9) is consistent. Meaning that, for all ε>0*

$$\lim_{n\to \infty }P(|\hat{I}\left(X;Y|Z\right)-I\left(X;Y|Z\right)\left|>\varepsilon \right)=0.$$

*In addition, $\hat{I}\left(X;Y|Z\right)$ is asymptotically unbiased, that is*

$$\lim_{n\to \infty }E\left[\hat{I}\left(X;Y|Z\right)-I\left(X;Y|Z\right)\right]=0.$$



**Proof.** It is well known that all linear combination of consistent estimators is consistent. This directly stems from Slutsky’s theorem [40]. It remains to show the consistency of each term in the right-hand side of Equation (Equation 9). Histogram-based estimators $\hat{H}\left(X^{\ell },Z^{\ell }\right)$, $\hat{H}\left(Y^{\ell },Z^{\ell }\right)$, $\hat{H}\left(X^{\ell },Y^{\ell },Z^{\ell }\right)$ and $\hat{H}\left(Z^{\ell }\right)$ are consistent according to [41]. By analogy, we only show the consistency of the estimator $\hat{H}\left(X^{t},Z^{t}|X^{\ell },Z^{\ell }\right)$, the same results apply to the remaining estimators. Let ε>0, we write
$$\begin{array}{cc} \; & P(|\hat{H}\left(X^{t},Z^{t}|X^{\ell },Z^{\ell }\right)-H\left(X^{t},Z^{t}|X^{\ell },Z^{\ell }\right)|>\varepsilon )\\ =\sum_{\begin{array}{c} x^{\ell }\in \Omega \left(X^{\ell }\right)\\ z^{\ell }\in \Omega \left(Z^{\ell }\right) \end{array}} & P(|\hat{H}\left(X^{t},Z^{t}|X^{\ell },Z^{\ell }\right)-H\left(X^{t},Z^{t}|X^{\ell },Z^{\ell }\right)|>\varepsilon |X^{\ell }=x^{\ell },Z^{\ell }=z^{\ell })\\ \times & P(X^{\ell }=x^{\ell },Z^{\ell }=z^{\ell }). \end{array}$$
Now, conditionally to given values of $X^{\ell }$ and $Z^{\ell }$, the estimator $\hat{H}\left(X^{t},Z^{t}|X^{\ell },Z^{\ell }\right)$ is the traditional *k*-nearest neighbors built using the maximum-norm distance. This estimator is shown to be consistent, the reader can refer to [42] for more details. In other words,
$$\lim_{n\to \infty }P(|\hat{H}\left(X^{t},Z^{t}|X^{\ell },Z^{\ell }\right)-H\left(X^{t},Z^{t}|X^{\ell },Z^{\ell }\right)\left|>\varepsilon \right|X^{\ell }=x^{\ell },Z^{\ell }=z^{\ell })=0.$$
This concludes the proof of consistency. Moreover, knowing that the histogram and *k*-nearest neighbors estimators are asymptotically unbiased, it is plain that our estimator also has this property.    □

### 3.2. Experimental Illustration

We compare in this section our estimator, CMIh, with several estimators described in Section 2, namely FP [12], MS [25], RAVK [24], and LH [16]. FP, MS and RAVK are methods based on the *k*-nearest neighbour approach. As for CMIh, the hyper-parameter *k* for these methods is set to the maximum value of ⌊n/10⌋ and 1, where *n* is the number of sampling points. To be consistent, we use for all three methods the widely used $\left(0\;-\;D_{\ell }\right)$ distance for the qualitative components: this distance is 0 for two equal qualitative values and $D_{\ell }$ otherwise. In our experiments, $D_{\ell }$ is set to 1, following [25]. Laslty, for FP, which was designed for quantitative data, we set the minimum value of $n_{FP,W,i}$ to 1 to avoid $n_{FP,W,i}=0$ in Equation (Equation 2). Moreover, LH is a histogram method based on MDL [16]. We use the default values for the hyper-parameters of this method: the maximum number of iterations, $i_{max}$, is set to 5, the threshold to detect qualitative points is also set to 5, the number of initial bins in quantitative component, $K_{init}$, is set to 20log(n) and the maximum number of bins, $K_{max}$, is set to 5log(n) (all entropies are computed in natural logarithm).

To assess the behaviour of the above methods, we first consider the mutual information with no conditioning (I(X;Y)), then with a conditioning variable which is independent of the process so that I(X;Y|Z)=I(X;Y), and finally with a conditioning variable which makes the two others independent, such that I(X;Y|Z)=0. We illustrate these three cases by either considering that *X* and *Y* are both quantitative or mixed, in which case they can have either balanced or unbalanced qualitative classes. Lastly, following [16,25], the conditioning variable *Z* is always qualitative.

Each (conditional) mutual information is computable theoretically so that one can measure the mean squared error (MSE) between the estimated value and the ground truth, which will be our evaluation measure. For each of the above experiments, we sample data with sample size *n* varying from 500 to 2000 and generate 100 data sets per sample size to compute statistics. More precisely, we use the following experimental settings, the first three ones being taken from [16,23,25]. The last four ones shed additional light on the different methods. Note that, as we reuse here the settings defined in [16,23,25], qualitative variables are generated either from a uniform distribution on a discrete set, a binomial distribution or a Poisson distribution, this latter case being an exception to our definition of what is a qualitative variable. We do not want to argue here on whether the Poisson variable should be considered quantitative or qualitative and simply reproduce here a setting used in previous studies for comparison purposes.

*MI quantitative.*$\left(\begin{array}{c} X\\ Y \end{array}\right)∼N\left(\left(\begin{array}{c} 0\\ 0 \end{array}\right),\left(\begin{array}{cc} 1 & 0.6\\ 0.6 & 1 \end{array}\right)\right)$ with $I\left(X;Y\right)=-\log\left(1-0.6^{2}\right)/2$.*MI mixed.*X∼U({0,…,4}) and Y|X=x∼U([x,x+2]), we get I(X;Y)=log(5)−4log(2)/5;*MI mixed imbalanced.*X∼Exp(1) and $Y|X=x∼0.15\delta_{0}+0.85Pois\left(x\right)$. The ground truth is $I\left(X;Y\right)=0.85\left(2\log2-\gamma -\sum_{k=1}^{\infty }\log k2^{-k}\right)\approx 0.256$, where γ is the Euler-Mascheroni constant.*CMI quantitative, CMI mixed and CMI mixed imbalanced.* We use the previous setting and add and independent qualitative random variable Z∼Bi(3,0.5).*CMI quantitative ⊥⊥.*Z∼Bi(9,0.5), X|Z=z∼N(z,1) and Y|Z=z∼N(z,1), the ground truth is then I(X;Y|Z)=0.*CMI mixed ⊥⊥.*Z∼U({0,…,4}), X|Z=z∼U([z,z+2]) and Y|Z=z∼Bi(z,0.5), the ground truth is then I(X;Y|Z)=0.*CMI mixed imbalanced ⊥⊥.*X∼Exp(10), Z|X=x∼Pois(x) and Y|Z=z∼Bi(z+5,0.5), the ground truth is I(X;Y|Z)=0.

Figure 1 displays the mean squared error (MSE) of the different methods in the different settings on a log-scale. As one can note, FP performs well in the purely quantitative case with no conditioning but is, however, not competitive in the mixed data case. MS and RAVK are close to each other and, not surprisingly, they have similar performance in most cases. MS, however, has a main drawback as it gives the value 0, or close to 0, to the estimator in some particular cases. Indeed, as noted by [25], if, for all points *i*, the *k*-nearest neighbour is always determined by *Z*, then, regardless of the relationship between *X*, *Y* and *Z*, $k_{MS,i}=n_{MS,XZ,i}=n_{MS,YZ,i}=n_{MS,Z,i}$ and the estimator equals to 0.

In addition, if the *k*-nearest-neighbour distance of a point *i*, $\rho_{k,i}/2$, is such that $\rho_{k,i}/2\geq D_{\ell }$ where $D_{\ell }\in N$ is the distance between different values of qualitative variables, then one has:$$n_{MS,YZ,i}=n_{MS,Z,i}=n\;\;and\;\;k_{MS,XYZ,i}=n_{MS,XZ,i}.$$
The first equality directly derives from the fact that one needs to consider points outside the qualitative class of point *i* (as $\rho_{k,i}/2\geq D_{\ell }$) and that all points outside this class are at the same distance ($D_{\ell }$). By definition, $n_{MS,YZ,i}\leq n_{MS,Z,i}$; furthermore, $n_{MS,Z,i}\leq n_{MS,YZ,i}$ as a neighbour of *i* in XZ with distance $\geq D_{\ell }$ is a neighbour of *i* in XYZ as *Y* cannot lead to a higher distance, which explains the second equality.

If a majority of points satisfy the above condition ($\rho_{k,i}/2\geq D_{\ell }$), then MS will yield an estimator close to 0, regardless of the relation between the different variables. This is exactly what is happening in the mixed and mixed imbalance cases as the number of nearest points considered, at least 50, can be larger than the number of points in a given qualitative class. In such cases, MS will tend to provide estimators close to 0, which is the desired behaviour in the bottom-middle and bottom-right plots of Figure 1, but not in the top-middle, top-right, middle-middle and middle-right plots (in these latter cases, the ground truth is not 0 which explains the relatively large MSE value of MS and RAVK). Our proposed estimator does not suffer from this drawback as we do not directly compare two different types of distances, one for quantitative and one for qualitative data.

Comparing LH and CMIh, one can see that, overall, these two methods are more robust than the other ones. The first and second lines of Figure 1 show that the additional independent qualitative variables *Z* does not have a large impact on the accuracy of the two estimators. The comparison of the second and third lines of Figure 1 furthermore suggests that, if the relationship between variables changes, the two estimators still have a stable performance.

*Sensitivity to dimensionality.* We conclude this comparison by testing how sensitive the different methods are to dimensionality. To do so, we first increase the dimensionality of the conditioning variable *Z* from 1 to 4 in a setting where *X* and *Y* are dependent and independent of *Z* (we refer to this setting as M-CMI for multidimensional conditional mutual information):$$X∼U\left(\left\{0,\ldots ,4\right\}\right),\;\;Y|X=x∼U\left(\left[x,x+2\right]\right),\;\;Z_{r}∼Bi\left(3,0.5\right),\;\;r\in \left\{0,\ldots ,4\right\}.$$
The ground truth in this case is $I\left(X;Y|Z_{1},\cdots ,Z_{4}\right)=I\left(X;Y\right)=\log\left(5\right)-4\log\left(2\right)/5$.

The results of this first experiment, based on 100 samples of size 2000 for the different components of *Z* (from 0 to 4), are displayed in Figure 2 (left). As one can observe, our method achieves an MSE close to 0.001 even though the dimension increases to 4. LH has a comparable accuracy for small dimensions but deviates from the true value for higher dimensions. For MS and RAVK, as mentioned in Mesner and Shalizi [25], when *X* and *Y* have fixed-dimension, the higher the dimension of *Z*, the greater the probability that the estimator will give a zero value. This can explain why for dimensions above 1, the MSE remains almost constant for these two methods. Lastly, FP performs poorly when increasing the dimension of the conditioning set.

It is also interesting to look at the computation time of each method on the above data, given in Table 1. One can note that our method is faster than the other ones and remains stable when the dimension of *Z* increases.

Let *B* denote the cardinal of the Cartesian product $X^{\ell }\times Y^{\ell }\times Z^{\ell }$ (B=1 when all variables are quantitative and B=4 in the setting retained here). The complexity of computing the four entropy terms in CMIh (Equation (Equation 9)) containing only qualitative variables is O(Bn) according to Equation (Equation 6). For the other entropy terms, one needs to apply at most *B* times the computation in Equation (Equation 8), which has an average complexity of $O({\left(\frac{n}{B}\right)}^{2}km_{t})$ (and $O(\left(\frac{n}{B}\right)\log\left(\frac{n}{B}\right)\left(k+m_{t}\right))$ for the approximation using KD-trees [43]), where $\frac{n}{B}$ represents the average number of sample points considered in Equation (Equation 8), k=max(⌊n/10⌋,1), and $m_{t}$ is the number of quantitative components over all variables ($m_{t}=2$ in the setting considered here). Thus, the overall complexity of CMIh is $O(Bn+\frac{km_{t}n^{2}}{B})$ (and $O(Bn+\left(k+m_{t}\right)n\log\left(\frac{n}{B}\right))$ with KD-trees). In contrast, the complexity of MS, RAVK and FP is $O(kmn^{2})$ (and O((k+m)nlog(n)) using KD-trees), where *m* is the number of dimensions over all variables (m=6 in the setting considered here). This explains the differences observed in Table 1. Lastly, note that the complexity of LH is $O(K_{max}^{m}K_{init}^{2}i_{max}m_{t})$ [16], which limits its application to very small datasets.

We then focus on the multivariate version of (unconditional) mutual information for mixed data based using the following generative process (this setting is referred to as M-MI for multidimensional mutual information):$$\left(\begin{array}{c} X_{1}\\ Y_{1} \end{array}\right)∼N\left(\left(\begin{array}{c} 0\\ 0 \end{array}\right),\left(\begin{array}{cc} 1 & 0.6\\ 0.6 & 1 \end{array}\right)\right),\;\;X_{2}∼U\left(\left\{0,\ldots ,4\right\}\right),\;\;Y_{2}|X_{2}=x_{2}∼U\left(\left[x_{2},x_{2}+2\right]\right),$$
$$X_{3}∼Exp\left(1\right)\;\;\mathrm{and}\;\;Y_{3}|X_{3}=x_{3}∼0.15\delta_{0}+0.85\mathrm{Pois}\left(x_{3}\right).$$
The ground truth in this case is $I(X_{1},X_{2},X_{3};Y_{1},Y_{2},Y_{3})\approx 1.534$.

Figure 2 (middle) displays the results obtained by the different methods but LH, computationally too complex to be used on datasets of a reasonable size, when the number of observations increases from 500 to 2000. As one can note, CMIh is the only method yielding an accurate estimate of the mutual information on this dataset. Both RAVK and MS suffer again from the fact that they yield estimates close to 0, which is problematic on this data. We give below another setting in which this behaviour is interesting; it remains nevertheless artificial.

Lastly, we consider the case where the two variables of interest are conditionally independent (we refer to this case as M-ICMI for multidimensional independent conditional mutual information). The generative process we used is:$$Z_{1}∼U\left(\left\{0,\ldots ,4\right\}\right),\;\;Z_{2}∼Bi\left(3,0.5\right),\;\;Z_{3}∼Exp\left(1\right),\;\;Z_{4}∼Exp\left(10\right),$$
$$X_{1},X_{2}\left|\left(Z_{3}=z_{3},Z_{4}=z_{4}\right)∼N\left(\left(\begin{array}{c} z_{3}\\ z_{4} \end{array}\right),\left(\begin{array}{cc} 1 & 0\\ 0 & 1 \end{array}\right)\right),\;\;X_{3}\right|\left(Z_{1}=z_{1},Z_{2}=z_{2}\right)∼Bi\left(z_{1}+z_{2},0.5\right),$$
$$Y|\left(Z_{1}=z_{1},Z_{2}=z_{2}\right)∼Bi\left(z_{1}+z_{2},0.5\right).$$
The ground truth in this case is $I(X_{1},\cdots ,X_{3};Y|Z_{1},\cdots ,Z_{4})=0$.

Figure 2 (right) displays the results obtained on all methods but LH. As for the univariate case, both RAVK and MS obtain very good results here but this is due to their pathological behaviour discussed above. CMIh yields a reasonable estimate (with an MSE below 0.1) when the number of observations exceeds 1250. FP fails here to provide a reasonable estimate.

Overall, CMIh, which can be seen as a trade-off between *k*-nearest neighbour and histogram methods, performs well, both in terms of the accuracy of the estimator and in terms of the time needed to compute this estimator. Among the pure *k*-nearest neighbour methods, MS, despite its limitations, remains the best one overall in our experiments in terms of accuracy. Its time complexity is similar to the ones of the other methods of the same class. The pure histogram method LH performs well in terms of accuracy of the estimator, but its computation time is prohibitive. Two methods thus stand out from our analysis, namely CMIh and MS.

## 4. Testing Conditional Independence

Once an estimator for mutual information has been computed, it is important to assess to which extent the obtained value is sufficiently different from or sufficiently close to 0 so as to conclude on the dependence or independence of the involved variables. To do so, one usually relies on statistical tests, among which permutation tests are widely adopted as they do not require any modelling assumption [38]. We also focus on such tests here which emulate the behaviour of the estimator under the null hypothesis (corresponding to independence) by permuting values of variables. Recently, Runge [39] showed that, for conditional tests and purely quantitative data, local permutations that break any possible dependence between *X* and *Y* while preserving the dependence between *X* and *Z* and between *Y* and *Z* are to be preferred over global permutations. Our contribution here is to extend this method to mixed data.

### 4.1. Local-Adaptive Permutation Test for Mixed Data

Let us consider a sample of independent realisations, denoted ${\left(X_{i},Y_{i},Z_{i}\right)}_{i=1,\ldots ,n}$, generated according to the distribution $P_{XYZ}$ where *X*, *Y* and *Z* are multidimensional variables with quantitative and/or qualitative components. From this sample, one can compute an estimator, denoted $\hat{I}\left(X;Y|Z\right)$, of the conditional mutual information using the hybrid method CMIh introduced in Section 3. In order to perform a permutation test, one needs to generate samples, under the null hypothesis, from the distribution $P_{X|Z}\left(x|z\right)P_{Y|Z}\left(y|z\right)P_{Z}\left(z\right)$. When the conditioning variable *Z* is qualitative, this boils down to randomly permuting the marginal sample of *X* while preserving the one of *Y*, conditionally to each possible value of *Z* [35]. In the quantitative case, one proceeds in a similar way and permutes the *X* values of the neighbours of each point *i* [35,39]. In our case, as the variable *Z* possibly contains quantitative and qualitative components, we propose to use an adaptive distance dist which corresponds to the absolute value if the component is quantitative and to the (0−∞) distance (which is 0 for identical values and *∞* for different values) if the component is qualitative. For $Z_{i}={\left(Z_{i}^{1},\ldots ,Z_{i}^{m}\right)}^{T}$ and $Z_{j}={\left(Z_{j}^{1},\ldots ,Z_{j}^{m}\right)}^{T}$ two realizations of the random vector *Z*, where *m* is the dimension of the data, the distance between these two points is then defined as:$$D\left(Z_{i},Z_{j}\right)=\max_{r\in \{1,\ldots ,m\}}dist\left(Z_{i}^{r},Z_{j}^{r}\right).$$
The neighbourhood of $Z_{i}$ consists in the set of *k* points closest to $Z_{i}$ according to *D*. Using the same *k* for all observations may, however, be problematic since it is possible that the $k^{th}$ closest point is at a distance *∞* of a given point $Z_{i}$ when *k* is large. In such a case, all points are in the neighbourhood of $Z_{i}$. To avoid this, we adapt *k* to each observation using one parameter $k_{i}$ for each observation $Z_{i}$: if *Z* is purely quantitative, then $k_{i}=k$, where *k* is a global hyper-parameter, otherwise $k_{i}=\min\left(k,n_{i}^{\ell }\right)$, where $n_{i}^{\ell }$ is the number of sample points which have the same qualitative values as $Z_{i}$.

Then, to generate a permuted sample, for each point *i* one permutes $X_{i}$ with the *X* value of a randomly chosen point in the neighbourhood of *i* while preserving $Y_{i}$ and $Z_{i}$: a permuted sample thus takes the form ${\left(X_{\pi (i)},Y_{i},Z_{i}\right)}_{i=1,\ldots ,n}$, where π(i) is a random permutation over the neighbourhood of *i*. By construction, a permuted sample is drawn under the null hypothesis since the possible conditional dependence is broken by the permutation. Many permuted samples finally are created, from which one can compute CMIh estimators under the null hypothesis. Comparing theses estimators to the one of the original sample allows one to determine whether the null hypothesis can be rejected or not [38]. Note that, in practice, the permutations are drawn with replacement [44]. The main steps of our local-adaptive permutation test are summarised in Algorithm 2.
**Algorithm 2** Local-Adaptive permutation test**Input**${\left(X_{i},Y_{i},Z_{i}\right)}_{1\leq i\leq n}$ the data, *B* the number of permutations, isCat indexes of qualitative component, *k* the hyper-parameter;Compute $\hat{I}\left(X,Y|Z\right)$ from the original data;Separate qualitative and quantitative components of *Z* as ${\left(Z_{i}^{t},Z_{i}^{\ell }\right)}_{1\leq i\leq n}$ using isCat;**for**i∈{1,…,n}**do**    **if** $Z^{\ell }\neq \emptyset$ **then**        $n_{i}^{\ell }=length\left(\left\{m\in \left\{1,\cdots ,n\right\}:Z_{m}^{\ell }=Z_{i}^{\ell }\right\}\right)$;        $k_{i}=\min\left(k,n_{i}^{\ell }\right)$;    **end if**    Compute $d_{i}^{k_{i}}$, the distance from $Z_{i}$ to its $k_{i}$-nearest-neighbour in *Z* by applying two different distances metrics, respectively, to two different types of components;    $N_{i}=\left\{j\in \left\{1,\cdots ,n\right\}:|\right|Z_{j}-Z_{i}\left|\right|\leq d_{i}^{k_{i}}\}$;**end for****for**b∈{1,…,B}**do**    Generate a sample ${\left(X_{\pi_{b}\left(i\right)},Y_{i},Z_{i}\right)}_{1\leq i\leq n}$ locally permuted with respect to ${\left(N_{i}\right)}_{1\leq i\leq n}$;    Compute the associated estimator $\hat{I}\left(X_{\pi_{b}},Y|Z\right)$;**end for**Estimate the *p*-value as
(10)$$\hat{p}=\frac{1}{B}\sum_{b=1}^{B}{𝟙}_{\hat{I}\left(X_{\pi_{b}},Y|Z\right)\geq \hat{I}\left(X,Y|Z\right)};$$**Output** The *p*-value $\hat{p}$.

### 4.2. Experimental Illustration

We first propose an extensive analysis on simulated data and then perform an analysis on a real world data set. We compare our test, denoted by LocAT, with two permutation tests: the first one is the local permutation test, denoted by LocT, designed initially for purely quantitative data proposed by Runge [39] and directly extended to mixed data using the (0−∞) distance for qualitative components; the second test is the global permutation test, denoted by GloT. For LocT and LocAT, we set the hyper-parameter $k_{perm}$ to 5 as proposed by Runge [39]. For all tests, we set the number of permutation, *B*, to 1000. We study the behaviour of each test with respect to the two best estimators highlighted in Section 3, CMIh and MS. It is important to note here that, in order to be consistent with the parameters of the original method, for MS we use the (0−1) distance in the qualitative component to compute the estimator rather than the (0−∞) distance as in the permutation method. We use rank transformation in each quantitative component which has the advantage of preserving the order and putting all quantitative components on the same scale (the “first” method is used to break potential ties).

#### 4.2.1. Simulated Data

We consider here that *X*, *Y* and *Z* are uni-dimensional but all estimators and independence tests can be used when the variables are multi-dimensional, as illustrated in the experiments conducted on the real dataset. We furthermore focus on three classical structures of causal networks: the chain (X→Z→Y), the fork (X←Z→Y), and the collider (X→Z←Y). For the chain and the fork, *X* and *Y* are dependent and independent conditionally to *Z*; for the collider, *X* and *Y* are independent and dependent conditionally to *Z*. In the sequel, the qualitative variables or components with infinite possible values are treated as quantitative ones. For each structure, one can potentially distinguishes eight configurations, depending on the type, quantitative (’*t*’) or qualitative (’*ℓ*’), of each of the three variables *X*, *Y* and *Z*. The configuration ’tℓt’ corresponds for example to the situation where *X* and *Z* are quantitative and *Y* qualitative. Note that as *X* and *Y* play a similar role, we only consider six cases. Details of the data generating process are provided in Appendix A. All samples contain 500 points.

The results obtained for each method, each structure and each configuration are reported in Table 2. For the chain and the fork, which are conditional independence structures, the acceptance rate corresponds to the percentage of the *p*-values that are above the thresholds 0.01 and 0.05 for 10 repetitions of each method in each configuration. For the collider, the acceptance rate corresponds to the percentage of the *p*-value that is under the thresholds 0.01 and 0.05 for 10 repetitions of each method in each configuration. In all cases, the closer the acceptance rate is to 1, the better.

As one can note, the global test does not perform well in the configurations ’ttt’ and ’ttℓ’ of the chain and fork structures, for both CMIh-GloT and MS-GloT. In addition, it does not perform well on the ’tℓt’ configuration of the chain and fork structures for CMIh. It nevertheless performs well for CMIh on the collider structure over all configurations, but not for MS. Overall, its global performance is relatively poor compared to the two local tests LocT and LocAT. The local test LocT performs relatively well on the chain and fork structures for all configurations but ’ttℓ’. It performs well for CMIh and the collider structure on all configurations but ’tℓℓ’; it does not, however, perform well for MS on this structure as only two configurations are correctly treated, ’ttt’ and ’ℓℓℓ’. Finally, the local adaptive test, LocAT, performs well on all configurations of all structures for CMIh. For MS, it performs well on the chain and fork structures but not on the collider structure where the results are identical to the ones obtained with the standard local test LocT. Note that the bad results obtained for all tests with MS on the collider structure are directly related to the limitations pointed out in the previous section. Indeed, the estimator given by MS on all configurations but ’ttt’ and ’ℓℓℓ’ is close to 0 as $\rho_{k,i}/2\geq D_{\ell }$ if there is at least one quantitative variable (due to rank transformation).

Overall, the combination CMIh with the test LocAT allows one to correctly identify the true (in)dependence relation on all configurations of all structures.

#### 4.2.2. Real Data

We consider here three real datasets to illustrate the behaviour of our proposed estimator and test. Given the performance of the global permutation test on the simulated data, we do not use it here and compare four estimator–test combinations: CMIh-LocT, CMIh-LocAT, MS-LocT and MS-LocAT.

##### Preprocessed DWD Dataset

This climate dataset was originally provided by the Deutscher Wetterdienst (DWD) and preprocessed by Mooij et al. [45]. It contains 6 variables (altitude, latitude, longitude, and annual mean values of sunshine duration over the years 1961–1990, temperature and precipitation) collected from 349 weather stations in Germany. We focus here on three variables, *latitude*, *longitude* and *temperature*, this latter variable being discretized into three balanced classes (low, medium and high) in order to create a mixed dataset. The goal here is to identify one unconditional independence (Case 1) and one conditional dependence (Case 2):Case 1: *latitude* is unconditionally independent of *longitude* as the 349 weather stations are distributed irregularly on the map.Case 2: *latitude* is dependent of *longitude* given *temperature* as both *latitude* and *longitude* act on *temperature*: moving a thermometer towards the equator will generally result in an increased temperature, and climate in West Germany is more oceanic and less continental than in East Germany.

The *p*-value for each method is shown in Table 3. For Case 1, the *p*-value should be high so that the null hypothesis is not rejected, whereas it should be small for Case 2 as the correct hypothesis is $H_{1}$. Note that as there is no conditional variable in Case 1, the permutation tests LocT and LocAT give the same results.

As one can note from Table 3, under both thresholds 0.01 and 0.05, CMIh-LocT and CMIh-LocAT succeed in giving the correct independent and dependent relations. In contrast, MS-LocT and MS-LocAT only identify the independent relation at the threshold 0.01 and never correctly identify the conditional dependency.

##### ADHD-200 Dataset

This dataset contains phenotypic data on kids with ADHD (Attention Deficit Hyperactivity Disorder) [46]. It contains 23 variables. We focus here on four variables: *gender*, *attention deficit level*, *hyperactivity/impulsivity level* and *medication status*, *gender* and *medication status* being binary categorical variables. The dataset contains 426 records after removing missing data. Following previous studies, we consider two independence relations:Case 1: *gender* is independent of *hyperactivity/impulsivity level* given *attention deficit level*, which has been confirmed by several studies [47,48].Case 2: *hyperactivity/impulsivity level* is independent of *medication status* given *attention deficit level*, which has been confirmed by Cui et al. [49].

The *p*-values obtained for the different estimator–test combinations are reported in Table 4. For this dataset, the *p*-values should be sufficiently high so that the null hypothesis is not rejected in both cases.

As one can note, regardless of whether the threshold is 0.01 or 0.05, all four methods reach the correct conclusion in both cases. CMIh-LocT and CMIh-LocAT have the same performance as the conditional variable is quantitative. From the previous simulated experiments we can reasonably infer that these two could perform better if more records were collected. MS-LocT and MS-LocAT give a *p*-value of 1 because the conditional mutual information in MS is 0; we observe here the same degenerate behaviour for this estimator as the one discussed in Section 3.

##### EasyVista IT Monitoring System

This dataset consists of five time series collected from an IT monitoring system with a one minute sampling rate provided by EasyVista (https://www.easyvista.com/fr/produits/servicenav, accessed on 31 August 2022). We focus on five variables: *message dispatcher* (activity of a process that orient messages to other process with respect to different types of messages), which is a quantitative variable, *metric insertion* (activity of insertion of data in a database), which is also a quantitative variable, *status metric extraction* (status of activity of extraction of metrics from messages), which is a qualitative variable with three classes, namely normal (≈75% of the observations), warning (≈20% of the observations) and critical (≈5% of the observations), *group history insertion* (activity of insertion of historical status in database), which is again a quantitative variable, and *collector monitoring information* (activity of updates in a given database) another quantitative variable. We know exact lags between variables, so we synchronise the data as a preprocessing step.

For this system we consider three cases:Case 1 represents a conditional independence between *message dispatcher* at time *t* and *metric insertion* at time *t* given *status metric extraction* at time *t* and *message dispatcher* and *metric insertion* at time t−1.Case 2 represents a conditional independence between *group history insertion* at time *t*, *collector monitoring information* at time *t* given *status metric extraction* at time *t* and *group history insertion* and *collector monitoring information* at time t−1.Case 3 represents a conditional dependence between *status metric extraction* at time *t* and *group history insertion* at time *t* given *status metric extraction* at time t−1.

For each case, we consider 12 datasets with 1000 observations each. The results, reported in Table 5, are based on the acceptance rates at thresholds 0.01 and 0.05 computed as in Section 4.2.1. Again, under each threshold, the closer the result is to 1, the better. Finally note that we conditioned on the past of each time series to eliminate the effect of the autocorrelation.

As one can see, CMIh-LocT and CMIh-LocAT yield exactly the same results on this dataset. Furthermore, the results obtained by these combinations are systematically better than the ones obtained when using MS as the estimator except for Case 2 with the threshold 0.05. However, on this case, all combinations correctly identify the conditional independence. Lastly, as before, MS yields poor results on Case 3, which corresponds to a collider structure. The explanation is the same as above for this structure and suggests that MS should not be used as an estimator to conditional mutual information.

Overall, the experiments on simulated and real datasets indicate that the combination CMIh-LocAT is robust to different structures and data types. This combination is well adapted to mixed data and provides the best results overall in our experiments.

## 5. Conclusions

We propose in this paper a novel hybrid method for estimating conditional mutual information in mixed data comprising both qualitative and quantitative variables. This method relies on two classical approaches to estimate conditional mutual information: *k*-nearest neighbour and histograms methods. A comparison of this hybrid method to previous ones illustrated its good behaviour, both in terms of accuracy of the estimator and in terms of the time required to compute it. We have furthermore proposed a local adaptive permutation test which allows one to accept or reject null hypotheses. This test is also particularly adapted to mixed data. Our experiments, conducted on both synthetic and real data sets, show that the combination of the hybrid estimator and the local adaptive test we have introduced is able, contrary to other combinations, to identify the correct conditional (in)dependence relations in a variety of cases involving mixed data. To the best of our knowledge, this combination is the first one fully adapted to mixed data. We believe that it will become a useful ingredient for researchers and practitioners for problems, including but not limited to (1) causal discovery where one aims to identify causal relations between variables of a given system by analyzing statistical properties of purely observational data, (2) graphical model inference where one aims to establish a graphical model which describes the statistical relationships between random variables and which can be used to compute the marginal distribution of one or several variables, and (3) feature selection where one aims to reduce the number of input variables by eliminating highly dependent ones.

## Figures and Tables

**Figure 1 entropy-24-01234-f001:**
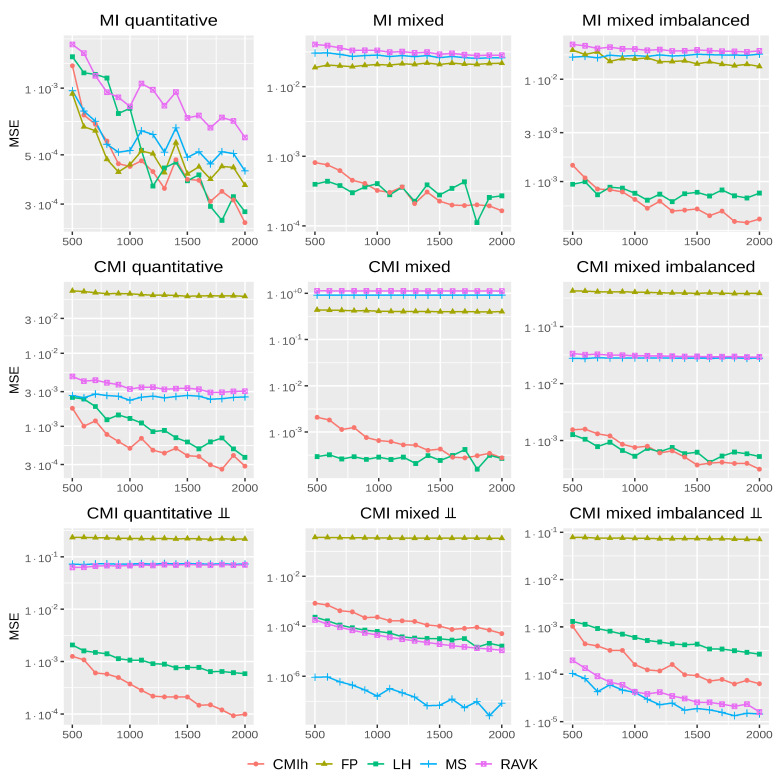
*Synthetic data with known ground truth.* MSE (on a log-scale) of each method with respect to the sample size (in abscissa) over the nine settings retained.

**Figure 2 entropy-24-01234-f002:**
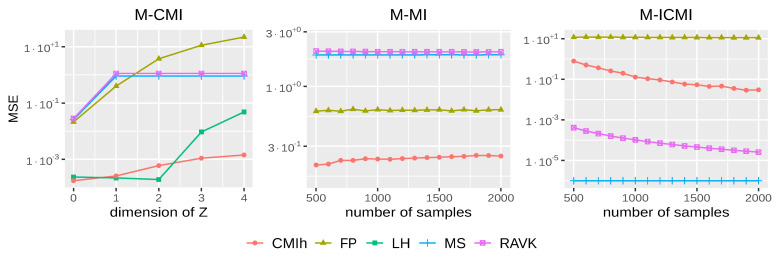
*Sensitivity to dimensionality***Left**: MSE (on a log-scale) of each method for the multidimensional conditional mutual information (M-CMI) when increasing the dimension (x-axis) of the conditional variable from 0 to 4; the sample size is fixed to 2000. **Middle**: MSE (on a log-scale) of each method but LH for the multidimensional mutual information (M-MI) when increasing the number of observations. **Right**: MSE (on a log-scale) of each method but LH for the multidimensional independent conditional mutual information (M-ICMI) when increasing the number of observations.

**Table 1 entropy-24-01234-t001:** We report, for each method, the mean computation time in seconds (its variance is given in parentheses), while varying the size of the conditional set from 0 to 4 with sample size fixed to 2000.

Dim of Z	0	1	2	3	4
CMIh	8.30(0.14)	5.30(0.05)	4.37(0.04)	4.16(0.04)	4.39(0.08)
FP	16.19(0.40)	22.09(0.27)	24.28(0.21)	25.91(0.08)	27.41(0.07)
LH	0.54(0.07)	1.09(0.02)	6.52(0.12)	58.58(13.74)	691.68(123.90)
MS	16.28(0.40)	22.08(0.07)	24.26(0.10)	26.07(0.06)	27.73(0.06)
RAVK	16.14(0.11)	22.07(0.07)	24.28(0.08)	25.89(0.09)	27.44(0.14)

**Table 2 entropy-24-01234-t002:** 0.01 and 0.05 threshold acceptance rates computed for the statistical test $H_{0}=X⊥\;\;\;⊥Y|Z$ versus H1=X⊥⊥Y|Z using the three tests LocT, LocAT and GloT on two estimators, CMIh and MS, on synthetic data. The number of sampling points is 500. Each acceptance rate is computed over 10 repetitions.

		CMIh-LocT		CMIh-LocAT		CMIh-GloT		MS-LocT		MS-LocAT		MS-GloT	
		0.01	0.05	0.01	0.05	0.01	0.05	0.01	0.05	0.01	0.05	0.01	0.05
	tℓt	1	1	1	1	0	0	1	1	1	1	1	1
	ttt	1	1	1	1	0	0	1	0.9	1	0.9	0	0
Chain	ℓℓt	1	0.9	1	0.9	1	0.8	1	1	1	1	1	1
	tℓℓ	1	1	1	1	1	1	1	1	1	1	1	1
	ttℓ	0	0	0.8	0.4	0	0	0	0	0.5	0.3	0	0
	ℓℓℓ	1	0.9	1	0.9	1	1	1	1	1	1	1	1
	tℓt	0.9	0.9	0.9	0.9	0	0	1	1	1	1	1	1
	ttt	1	1	1	1	0	0	1	1	1	1	0	0
	ℓℓt	1	1	1	1	1	1	1	1	1	1	1	1
Fork	tℓℓ	1	1	1	0.9	1	1	1	1	1	1	1	1
	ttℓ	0	0	0.9	0.8	0	0	0	0	0.8	0.5	0	0
	ℓℓℓ	1	1	1	1	1	1	1	0.9	1	1	1	1
	tℓt	1	1	1	1	1	1	0	0	0	0	0	0
	ttt	1	1	1	1	0.8	0.9	1	1	1	1	1	1
	ℓℓt	1	1	1	1	1	1	0	0	0	0	0	0
Collider	tℓℓ	0	0	0.4	0.7	0	0	0	0	0	0	0	0
	ttℓ	0.6	1	1	1	0.2	0.4	0	0	0	0	0	0
	ℓℓℓ	1	1	1	1	1	1	1	1	1	1	0.4	0.9

**Table 3 entropy-24-01234-t003:** DWD: *p*-values for the different estimator–test combinations of the statistical test, which is $H_{0}=X⊥\;\;\;⊥Y$ versus H1=X⊥⊥Y for Case 1, where *X* and *Y* correspond to *latitude* and *longitude*, and $H_{0}=X⊥\;\;\;⊥Y|Z$ versus H1=X⊥⊥Y|Z for Case 2, where *X*, *Y* and *Z* correspond to *latitude*, *longitude* and *temperature*. The number of sampling points is 349.

	CMIh-LocT	CMIh-LocAT	MS-LocT	MS-LocAT
Case 1	0.05	0.05	0.03	0.03
Case 2	0	0	0.09	0.08

**Table 4 entropy-24-01234-t004:** ADHD-200: *p*-values for the different estimator–test combinations of the statistical test $H_{0}=X⊥\;\;\;⊥Y|Z$ versus H1=X⊥⊥Y|Z where *X*, *Y* and *Z* correspond to *gender*, *hyperactivity/impulsivity level* and *attention deficit level* for Case 1 and *hyperactivity/impulsivity level*, *medication status* and *attention deficit level* for Case 2. The number of sampling points is 426.

	CMIh-LocT	CMIh-LocAT	MS-LocT	MS-LocAT
Case 1	0.36	0.36	1	1
Case 2	0.17	0.19	1	1

**Table 5 entropy-24-01234-t005:** EasyVista: 0.01 and 0.05 threshold acceptance rates for the different estimator–test combinations computed for the statistical test $H_{0}=X⊥\;\;\;⊥Y|Z$ versus $H_{1}$ = X⊥⊥Y|Z, where *X*, *Y* and *Z* correspond to ${\mathrm{message dispatcher}}_{t}$, ${\mathrm{metric insertion}}_{t}$ and the vector $\left({\mathrm{status metric extraction}}_{t},{\mathrm{message dispatcher}}_{t-1},{\mathrm{metric insertion}}_{t-1}\right)$ for Case 1, to ${\mathrm{group history insertion}}_{t}$, ${\mathrm{collector monitoring information}}_{t}$ and the vector $\left({\mathrm{status metric extraction}}_{t},{\mathrm{group history insertion}}_{t-1},{\mathrm{collector monitoring information}}_{t-1}\right)$ for Case 2 and ${\mathrm{status metric extraction}}_{t}$, ${\mathrm{group history insertion}}_{t}$ and ${\mathrm{status metric extraction}}_{t-1}$ for Case 3. The number of sampling points is 1000. Each acceptance rate is computed over 12 datasets of the same structure.

	CMIh-LocT		CMIh-LocAT		MS-LocT		MS-LocAT	
	0.01	0.05	0.01	0.05	0.01	0.05	0.01	0.05
Case 1	1	0.75	1	0.75	0.67	0.58	0.75	0.58
Case 2	1	0.67	1	0.67	0.92	0.75	1	0.83
Case 3	0.75	0.83	0.75	0.83	0	0	0	0

## Data Availability

All data presented in this study are publicly available: Simulated data are available at https://github.com/leizan/CMIh2022 (accessed on 25 August 2022); Preprocessed DWD dataset are available at https://webdav.tuebingen.mpg.de/cause-effect/ (accessed on 22 August 2022); ADHD-200 data are available at http://fcon_1000.projects.nitrc.org/indi/adhd200/index.html (accessed on 21 August 2022); IT monitoring data are available at https://easyvista2015-my.sharepoint.com/:f:/g/personal/aait-bachir_easyvista_com/ElLiNpfCkO1JgglQcrBPP9IBxBXzaINrM5f0ILz6wbgoEQ?e=OBTsUY (accessed on 1 July 2022).

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
