# Peer review of "A Conditional Mutual Information Estimator for Mixed Data and an Associated Conditional Independence Test"

_entropy, 2022, doi:10.3390/e24091234_

Round 1

Reviewer 2 Report

I find the work well written and well organized. The simulation study is well structured and comprehensive. The part of application to real data, on the other hand, is in my opinion small and poor; it should be expanded and discussed better. The proposed real datasets are all in the same applocation domain and time series type, while the simulations all involve cross sectional datasets.  Does the correlation over time of observations that are therefore not i.i.d. have no effect on the results? When the authors talk about discrete data, do they always mean numeric data, or could the method also work for categorical non ordinal variables (mixed type data generally means ordinal or binary variables as well)? The authors should make an effort to identify areas of application where their proposal might be relevant and useful and suggest them, also to enrich the conclusions and impacts of their proposal.

Round 2

Reviewer 1 Report

The revised version is looking good.

Reviewer 2 Report

The authors responded promptly to the comments I had made.  The manuscript has greatly improved and can now be accepted for publication